Chemotaxis and degradation of organophosphate compound by a novel moderately thermo-halo tolerant Pseudomonas sp. strain BUR11: evidence for possible existence of two pathways for degradation

Pailan Santanu
Saha Pradipta psaha@microbio.buruniv.ac.in
Microbiology Department, The University of Burdwan , West Bengal , India
Arora Pankaj
Electronic publication date: 2015 Nov 10
Publication date: 2015
Volume: 3
Electronic Location ID: e1378
Received 2015 Jul 8; Accepted 2015 Oct 14
Copyright: © 2015 Pailan and Saha
Copyright year: 2015
Copyright holder: Pailan and Saha
License: This is an open access article distributed under the terms of the Creative Commons Attribution License, which permits unrestricted use, distribution, reproduction and adaptation in any medium and for any purpose provided that it is properly attributed. For attribution, the original author(s), title, publication source (PeerJ) and either DOI or URL of the article must be cited.
License URL: https://creativecommons.org/licenses/by/4.0/

Keywords: Organophosphate insecticide, Parathion, Chemotaxis, 16S rRNA gene, Degradation

Funding: SERB, New Delhi SR/FT/LS-109/2010 State fellowship (Government of West Bengal) through University of Burdwan Work carried out for this manuscript was financially supported by SERB, New Delhi (through project no. SR/FT/LS-109/2010). Santanu Pailan is supported by a State fellowship (Government of West Bengal) through University of Burdwan. The funders had no role in study design, data collection and analysis, decision to publish, or preparation of the manuscript.

==============================
An organophosphate (OP) degrading chemotactic bacterial strain BUR11 isolated from an agricultural field was identified as a member of Pseudomonas genus on the basis of its 16S rRNA gene sequence. The strain could utilize parathion, chlorpyrifos and their major hydrolytic intermediates as sole source of carbon for its growth and exhibited positive chemotactic response towards most of them. Optimum concentration of parathion for its growth was recorded to be 200 ppm and 62% of which was degraded within 96 h at 37 °C. Growth studies indicated the strain to be moderately thermo-halo tolerant in nature. Investigation based on identification of intermediates of parathion degradation by thin layer chromatography (TLC), high performance liquid chromatography (HPLC), gas chromatography (GC) and liquid chromatography mass spectrometry (LC-MS/MS) provided evidence for possible existence of two pathways. The first pathway proceeds via 4-nitrophenol (4-NP) while the second proceeds through formation of 4-aminoparathion (4-APar), 4-aminophenol (4-AP) and parabenzoquinone (PBQ). This is the first report of chemotaxis towards organophosphate compound by a thermo-halo tolerant bacterium.

Introduction

Organophosphate (OP) compounds are extensively used throughout the world as insecticides, nematicide as well as chemical warfare agents (Kanekar, Bhadbhade & Deshpande, 2004; Yang et al., 2006). These compounds are toxic to human and other animals since they inhibit acetylcholine esterase and are known to disrupt normal functions of central nervous system followed by severe muscle paralysis and death (Shen et al., 2010). Although banned in many countries, several of them, especially parathion and methyl parathion are still being used indiscriminately in India for controlling insect pests of major crops like paddy, potato, mustard, cotton and vegetables (Pakala et al., 2007; Banerjee et al., 2014). These toxic compounds and their hydrolysis products remain in soil, air, surface water as well as ground water long after their application in the field (KaviKarunya & Reetha, 2012) and thus create health hazards. OP was first introduced as insecticide in the year 1947 and their half life varies (60–120 days for chlorpyrifos and 30–180 days for parathion) depending on soil pH, other abiotic and biotic factors (Singh & Walker, 2006; Gao et al., 2012). Chlorpyrifos was reported to be degraded by Bacillus pumilus more efficiently (80%) at basic and neutral pH compared to acidic pH (only 50%) (Anwar et al., 2009). However, Chen et al. (2012a) reported more efficient degradation at acidic pH by a fungus, Cladosporium cladosporioides Hu-01. OP compounds are acutely toxic to mammals, for example the LD50 of parathion for mammals is in the range 2–10 mg kg−1 (Singh & Walker, 2006).

Reports in the literature suggest that several soil microorganisms, such as Flavobacterium sp. ATCC 27551 (Sethunathan & Yoshida, 1973); Bacillus sp. and Pseudomonas sp. (Siddaramppa, Rajaram & Sethunathan, 1973); Pseudomonas diminuta-MG (Serdar et al., 1982; Mulbry et al., 1986); Pseudomonas putida KT2442 (Walker & Keasling, 2002); Pseudomonas sp. (Min-Kyeong et al., 2009), Enterobacter strain B-14 (Singh et al., 2004), Alcaligens faecalis DSP3 (Yang et al., 2005), Stenotrophomonas sp. strain YC-1 (Yang et al., 2006), Sphingomonas strain Dsp-2 (Li, He & Li, 2007), Paracoccus sp. strain TRP (Xu et al., 2008), B. pumilus strain C2A1 (Anwar et al., 2009) among bacteria and Verticillium sp. strain DSP (Fang et al., 2008), Acremonium sp. strain GFRC-1 (Kulshrestha & Kumari, 2011) and Cladosporium cladosporioides Hu-01 Chen et al., 2012a) among fungi are reported to be capable of degrading OP-compounds either co-metabolically and in a very few cases catabolically. So far three pathways by which parathion is degraded and the end products are channelized to TCA cycle have been reported (Singh & Walker, 2006). The most common pathway of parathion degradation is through formation of 4-NP (which is further degraded through formation of either 4-nitrocatechol (4-NC) and or PBQ) while the second pathway is through paraoxon and 4-NP and the third one is through formation of 4-APar and 4-AP (Munnecke & Hsies, 1976). The third pathway has been reported only for mixed bacterial culture and is believed to operate in limited oxygen environment (Munnecke & Hsies, 1976).

The first step in biodegradation is the bioavailability of a compound to the bacterial cells which may be accomplished by chemotaxis. In order to biodegrade, bacteria must have access to the target compounds either by dissolution of the target compounds in the aqueous phase or by adhesion of the bacteria directly to the non-aqueous phase liquid water interface. Survey of literature revealed reports of many bacteria that showed chemotactic activity towards xenobiotic compounds which have been postulated to be responsible for their efficient degradation (Pandey & Jain, 2002). However, extensive reports on positive chemotactic response towards OP compounds and some of their degradation intermediates such as 4-AP and 3,5,6 trichloro-2-pyridinol (TCP) are lacking in literature.

In this study we report for the first time positive chemotactic response of a novel, moderately thermo-halo tolerant bacterium Pseudomonas sp. strain BUR11 towards OP compound and some of their degradation intermediates such as 4-AP, TCP and 4-NP. We also demonstrate evidence for possible existence of two pathways for degradation of parathion in this newly isolated bacterial strain.

Materials and Methods

Chemicals

Parathion and its degradation intermediates (such as 4-NP, 4-AP, 4-NC, hydroquinone (HQ), PBQ and 1,2,4 benzenetriol (BT)) as well as TCP were purchased from Sigma. Chlorpyrifos was purchased from Dr. Ehrenstorfer GmbH (Merck, Germany). Solvents used in this study were of HPLC grade. Microbiological media, components, general chemicals were of highest standard and purchased from HiMedia, India.

Isolation of OP degrading bacteria

Soil samples were collected in sterile whirl pack bags (HiMedia, India) from an agricultural field in Burdwan (lat. 23.2500 °N and long. 87.8500 °E), West Bengal, India. For isolation of parathion utilizing bacteria, an enrichment culture was set up in minimal medium (MM) ((g L−1): K2HPO4, 0.2; KH2PO4, 0.8; MgSO47H2O, 0.2; CaSO4H2O, 0.1; NaMoO42H2O, 0.003; FeSO47H2O, 0.005; (NH4)2SO4, 1.0) supplemented with 200 ppm (mg L−1) parathion. One gram of soil sample was added to the 25 mL of sterile MM (in a 100 mL Erlenmeyer flask) supplemented with filter sterilized parathion as the sole source of carbon and energy and was incubated in an orbital shaker (120 rpm) at 37 °C for 1 month. After 1 month, 1 mL of inoculum was transferred to 25 mL of fresh MM with 200 ppm of parathion and incubated for 15 days and the process was repeated three more times. Dilution plating of this enrichment culture on MM agar (1.5% agar) plate supplemented with 200 ppm of parathion and incubation at 37 °C resulted in many isolated colonies which were purified by restreaking on fresh medium. Based on growth response on MM-parathion plate, one isolate designated BUR11 was selected for further studies.

Characterization and identification of isolate BUR11

Phenotypic characterization of strain BUR11 was carried out essentially according to Smibert & Krieg (1994). Growth at various temperatures (20 °C–55 °C) and NaCl concentrations was determined by using basal tryptone soya broth with varying NaCl concentrations (0.5–8.5%), followed by its periodic monitoring at 600 nm using a spectrophotometer. Strain BUR11 was also identified by an automated microbial identification system, VITEK 2 (BioMérieux, Inc., Hazelwood, MO, USA). Genomic DNA was isolated by Marmur’s protocol (Johnson, 1994); amplification of 16S rRNA gene and its sequencing was carried out according to Reddy et al. (2000). Determination of phylogenetic neighbors and pair wise 16S rRNA gene sequence identity was carried out at EzTaxon server (Chun et al., 2007). Phylogenetic tree was constructed according to published methods (Saha & Chakrabarti, 2006) with sequences of close phylogenetic neighbors, retrieved from EzTaxon server.

Growth studies and chemotactic behavior of strain BUR11

The ability of the stain BUR11 to grow on parathion, chlorpyrifos, 4-NP, PBQ, HQ, 4-AP, TCP and BT as sole sources of carbon was determined on MM broth supplemented with 200 ppm of the compounds and incubation at 37 °C with shaking at 120 rpm. Growth was monitored spectrophotometrically (Varion, Cary 50 Bio) at 600 nm, at regular time intervals.

The chemotactic activity of the strain BUR11 towards parathion, chlorpyrifos, TCP, 4-NP and 4-AP were investigated qualitatively (by drop plate and swarm plate assays) and quantitatively by capillary assay following established methods (Pandey et al., 2002; Arora & Bae, 2014). For drop plate assay, culture was grown on MM supplemented with respective chemoattractants, harvested (OD600 ∼ 0.7) by centrifugation (6,000g for 10 min) washed twice with phosphate buffered saline (PBS) and resuspended in assay medium (MM with 0.3% agarose), mixed well and poured onto 60 mm Petri plates. The test compounds (either as crystals or as solution) were placed in the centre of the plates which were then incubated at 37 °C. Swarm plate assay was carried out on assay medium (MM with respective toxic compound plus 0.3% agarose) with 60–75 µL of cells suspension in chemotaxis buffer, loaded centrally (in an agar cup that was cut aseptically, using sterile cork borer), in a 60 mm Petri plate. Appearance of bacterial growth in ring pattern (concentric rings in case of drop plate and exocentric in case of swarm plate assay) were recorded as positive chemotactic response. For quantitative capillary assay, the optimum concentration for each of the tested toxic compounds was determined by carrying out chemotaxis assay at various concentrations (from 50 to 500 ppm). For the assay, 10 µL graduated glass capillary tubes were filled with parathion, chlorpyrifos, TCP, 4-NP and 4-AP (in chemotaxis buffer) separately, and the suction end was sealed by sterile agarose gel. Each of these capillary tubes was inserted into a micro centrifuge tube (separately) containing a suspension (108 cells mL−1) of strain BUR11 cells and was incubated at 25 °C for 30 min. The contents of the capillary tubes were then serially diluted and plated onto non-selective tryptone soya agar plate, followed by determination of colony forming units (CFUs count) after overnight incubation at 37 °C. The strength of chemotactic response was expressed in terms of the chemotaxis index (CI), which is the ratio of the number of CFUs produced from the capillary containing the test compound(s) to CFUs produced from a control capillary (i.e., only the chemotaxis buffer without any chemo tactic compound). Citrate was used as the positive control.

Inoculum preparation and extraction of samples for degradation studies

In order to carry out biodegradation studies, strain BUR11 was grown on tryptone soya broth for overnight, cells were harvested by centrifugation, washed thrice with PBS (pH-7.0) to remove traces of medium and resuspended in 1.5 mL of MM. Cell suspension (0.5 OD corresponding to 6.5 × 107 cells mL−1) was inoculated into MM supplemented with parathion (200 ppm). Control flask was prepared exactly the same way without inoculum and all flasks were incubated at 37 °C with shaking at 120 rpm. Entire broth culture was centrifuged and the supernatant was used for extraction and amount of residual parathion in the medium was determined by HPLC after 0, 24, 48, 72 and 96 h of incubation.

For determination of hydrolysis products, after a periodic time interval of growth on MM-parathion, cell free culture supernatant was collected and was subjected to extraction (twice) using equal volume of ethyl acetate, followed by dehydration using anhydrous sodium sulphate (Na2SO4) and was dried by a rotary evaporator. The dried sample was dissolved in appropriate volume of acetonitrile, filtered through sterile disposable syringe filter (Millipore, 0.22 µm) and subjected to separation by TLC, LC-MS/MS and GC followed by library search for identification of metabolites.

Analytical methods

Quantification of residual parathion and intermediate metabolites by HPLC

Parathion and its degradation intermediates were quantified by HPLC using a HiQ sil C18 column (250 mm × 4.6 ID) and a Waters 515 HPLC equipped with a 486 tunable UV/Vis detector. Parathion and intermediate metabolites were detected at 274 nm and acetonitrile: water (80:20) was used as the mobile phase at a flow rate of 1 mL min−1. Compounds were identified by comparing their retention time to those of authentic standards. Quantitative aspects of parathion and other intermediates were calculated from their standard curves (Shen et al., 2010; Pailan et al., 2015).

Identification of metabolites by TLC, HPLC, LC-MS/MS and GC

For preliminary identification of hydrolytic intermediates during parathion hydrolysis, TLC was performed using precoated silica gel 60 F254 plates (20 × 20 cm; Merck). 10 µL of extracted sample was loaded on TLC Plate and was developed using n-hexane: acetone: ethyl acetate (80:10:10) solvent system. Visualization of TLC plate was carried out under UV light. HPLC was carried out as mentioned above and compounds were identified by comparing retention time of the test samples to that for authentic standards. LC-MS/MS was carried out using Sampler model no. G1329B for LC and MSQQQ Mass spectrometer for MS/MS (Agilent Technologies). The injection volume was 10 µL, solvent composition (methanol: ammonium formate (5 mM) 90:10) was used as mobile phase with flow rate of 0.5 mL min−1 and stop time was 15 min. MSMS was operated in following conditions, ion source was AJS-ESI, stop time was 25 min, gas temperature 325 °C, gas flow 6 L min−1 and capillary voltage were 4,000 V and 3,000 V. GC (Agilent Technologies), equipped with PHENOMENEX_ ZB_5_MS (30 m × 250 µm × 0.25 µm) column, was used for the separation of metabolites. Helium (He) and Nitrogen (N2) were used as quench gas and collision gas respectively with flow rate of 2.25 mL min−1 and 1.5 mL min−1 respectively. The injection volume was 2 µL. The identification of intermediate compounds was carried out by comparing their mass spectrum profiles either to that of NIST (NIST/EPA/NIH Mass spectral Library) library or to the same for standards.

Results

Isolation, characterization and identification of parathion degrading bacterium

Several bacterial strains capable of utilizing parathion as sole source of carbon for their growth were isolated from soil of an agricultural field by enrichment culture technique. One of them designated as BUR11 showed comparatively good growth response in MM-parathion broth and MM-parathion-agar plate and was selected for further study.

The isolate BUR11 was Gram negative, rod-shaped, motile and possessed nitrate and nitrite reductase activities. The strain could grow between 20 °C to 50 °C temperatures and could tolerate up to 8% NaCl (Fig. S1). Phenotypic characteristics of the strain are summarized in Table 1. The phylogenetic position of the strain is shown in Fig. 1. Although, it showed high value (99.78%) of 16S rDNA sequence similarity to the type strain of Pseudomonas aeruginosa (JCM 5962T), in absence of detail polyphasic taxonomic characterization, the strain BUR11 was identified as Pseudomonas sp. The strain has been deposited in Microbial Culture Collection, National Centre for Cell Science, Pune, India (accession number MCC 2328). Nucleotide sequence of the 16S rRNA gene for the strain BUR11 has been deposited in GenBank (accession number KF887018).

Figure 1 Phylogenetic tree.

Phylogenetic relationship between strain BUR11 and other closely related species of the genus Pseudomonas based on 16S rRNA gene sequences. The tree was generated by Neighbour-Joining method using TREECON software. Bootstrap values (as real value of 100 replications) are shown at the nodes. Bar, 0.05 base substitutions per site. Sequence from Rhizobacter dauci was used as an out-group.

Table 1 Phenotypic characteristics of the strain Pseudomonas sp. BUR11.

Test	Results	
Caseinase, Catalase, Oxidase, Phosphatase activity	+	
NaCl tolerance	Up to 8%	
Growth temperature	20 °C–50 °C, (37 °C optimum)	
Hydrolysis of Esculin, Gelatin	+	
*Utilization of L-Arabinose, Citrate, Galactose, Glucose, Malonate, D-Mannose, Xylitol, Xylose,	+	
Notes.

(+, positive reaction).

* Data taken from GN card of VITEK 2 System Version: 06.01.

Growth studies and chemotactic response of strain BUR11

Study of growth pattern revealed that the Pseudomonas sp. strain BUR11 could utilize parathion, chlorpyrifos (Fig. 2A) and their hydrolytic intermediates (such as 4-NP, 4-AP, HQ, PBQ, BT and TCP; Fig. 2B) as sole source of carbon. The strain could tolerate up to 500 ppm of parathion and chlorpyrifos while its optimum growth was recorded at 200 ppm for both the OP compounds. The ability of this motile strain BUR11, to utilize OP compounds and their hydrolysis intermediates prompted us to look for its chemotactic response towards these compounds.

Figure 2 Growth of BUR11 in presence of OP compounds and their hydrolytic intermediates.

Growth profile of strain BUR11 on (A) different OP compounds (as sole carbon source; 200 ppm). (B) On different hydrolysis intermediates of OP compounds (4-NP, HQ, PBQ, BT, 4-AP and TCP), citrate (0.5%) and on tryptone soya broth (3%).

Results from qualitative (Figs. S2A and S2B; drop plate and swarm plate assay respectively) as well as quantitative (Fig. 3) chemotaxis assays indicated positive chemotaxis for five compounds (parathion, 4-NP, 4-AP, chlorpyrifos and TCP) that were utilized as sole source of carbon for growth by the strain BUR11. The capillary chemotaxis assay indicated concentration dependent chemotaxis. As shown in Fig. 3, the CI values for all five compounds gradually increased with increasing concentrations until the optimal concentrations. Further increase in concentration led to sharp declines for chlorpyrifos, TCP and 4-AP while plateaus for parathion and 4-NP in the strength of the chemotactic response. The optimal chemotactic response concentrations were documented in the range between 200–250 ppm for all the compounds used as sole carbon source. The strongest chemotactic response was observed for chlorpyrifos and TCP, with CI values of 30.1 and 28.5 respectively, at their respective optimal response concentrations whereas CI of 16.25, 18.83 and 24.54 were recorded for parathion, 4-NP and 4-AP. To, the best of our knowledge, this is the first report of chemotaxis towards chlorpyrifos, parathion, 4-AP and TCP, by a moderately thermo-halo tolerant bacterium.

Figure 3 Quantitative capillary assay.

Quantitation of the chemotactic response and determination of optimal response concentration for BUR11 chemotaxis towards different test compounds using capillary assays.

Degradation, hydrolytic intermediates and pathways for parathion degradation by BUR11

Study of the degradation of parathion by Pseudomonas sp. strain BUR11 revealed that the strain could degrade 35% of parathion within 24 h and 62% of the same within 96 h as sole source of carbon (Fig. 4A). As evident from Fig. 4B, within first 24 h, there was increase in concentration of 4-NP, up to 30 ppm and this does not change until 72 h after which its concentration decreased rapidly, indicating its mineralization. Other intermediates (HQ, PBQ and BT) were detected in very low concentration, indicating their rapid conversion into intermediates which were channelized finally into TCA cycle.

Figure 4 Parathion degradation profile of BUR11.

(A) Parathion degradation profile by the strain BUR11 and (B) Fate of intermediates during parathion degradation by the strain BUR11.

Preliminary analysis by TLC (Fig. S3), indicated presence of three metabolites namely, 4-NP, PBQ, BT as major intermediates of hydrolysis. At least three more unidentified spots were detected during TLC analysis. From comparative analyses (by HPLC) of retention time (RT) for peaks of sample extracts and standards, HQ (RT of 2.258 min); BT (RT of 2.195 min), 4-NP (RT of 2.440 min) and PBQ (RT of 2.633 min) were detected as major intermediates of hydrolysis of parathion (Fig. S4). Separation of metabolites by GC followed by library search, detected and identified 4-APar (RT 19.029 min) in the parathion grown culture extract (Fig. S5). In LC-MS/MS analysis, presence of characteristic mass spectra for 4-NP (ion, m/z 137.9 > 108.0 > 92.0), HQ (ion, m/z 110 > 82.0 > 54.0 > 53.0), PBQ (ion, m/z 104.8), 4-AP (ion, m/z 104.9), BT (148.9 > 121.1 > 92.7) and 4-NC (ion, m/z 148.9 > 92.7) were detected (Fig. S6).

LC-MS/MS analysis thus confirmed the presence of six major hydrolysis intermediates, identified as 4-NP, HQ, PBQ, 4-NC, BT and 4-AP; while GC-screening predicted presence of 4-APar. Since both 4-NP (as well as 4-NC, BT, PBQ, HQ) and 4-APar (as well as 4-AP) were detected, we believe that in strain BUR11, possibly, two pathways for degradation of parathion exists. Thus, based on detection of these intermediates we propose possible pathways of degradation of parathion for the strain BUR11 (Fig. 5). The proposed first pathway proceeds via 4-NP that further degrades via two routes; one proceeds via 4-NC and BT whereas the other proceeds via PBQ and HQ. While the second proposed pathway operates through the formation of 4-APar, 4-AP and PBQ (Fig. 5).

Figure 5 Parathion degradation pathway.

Proposed pathway for the degradation of parathion by the strain BUR11.

Discussion

A parathion degrading bacterium, Pseudomonas sp. strain BUR11, was isolated from agricultural soil by enrichment culture technique. The strain BUR11 not only utilized parathion but also utilized its different hydrolytic intermediates like 4-NP, 4-AP, HQ, PBQ, BT as well as chlorpyrifos and its major hydrolytic intermediate, TCP as sole source of carbon for its growth. The strain could degrade 62% parathion within 96 h at 37 °C. Previously, Singh & Walker (2006) has reported that OP utilization by different species of Pseudomonas as sole source of carbon but most of the authors have reported Pseudomonas aeruginosa as a member of a mixed bacterial culture for utilization of parathion (Munnecke & Hsies, 1974; Pino, Dominguez & Penuela, 2011). Degradation of parathion by reported bacterial strains proceeds through formation of 4-NP only and no bacterial strain has been reported where two different pathways are operative for degradation of parathion.

Evidence based on TLC, HPLC, GC and LC-MS/MS techniques indicated evidence for possible existence of two pathways for degradation of parathion by the strain BUR11. To the best of our knowledge this is the first report of a bacterium with two possible pathways for degradation of parathion. In the first pathway, the strain BUR11 converts parathion to 4-NP. The proposed second pathway of degradation of parathion in BUR11 proceeds through formation of 4-APar and 4-AP (Fig. 5). To date, this second pathway is reported for a mixed bacterial culture that occurs under limited O2 condition (Munnecke & Hsies, 1976). Sharmila, Ramanand & Sethunathan (1989) reported a Bacillus sp. that could degrade methyl parathion, through nitro group reduction, in the presence of yeast extract. This second pathway of degradation of parathion involves reduction of nitro group to amine group, followed by hydrolysis of ester bond to produce 4-AP. Reductive transformation of parathion and methyl parathion to 4-APar, under aerobic condition, catalyzed by oxygen insensitive nitroreductase enzyme has been reported for Bacillus sp. (Yang et al., 2007). Similar light dependent aerobic reduction has been reported earlier for Anabaena sp. PCC7120 (Barton et al., 2004). Although not evaluated by enzyme assays, the strain BUR11, might possess similar O2 insensitive nitroreductase enzyme system to carry out conversion of parathion to 4-APar (i.e., nitro to amine group), to reduce toxicity, the ester bond is further hydrolyzed to release 4-AP, mineralized via PBQ and HQ. Moreover, the ability of the strain BUR11 to utilize all the intermediates (except 4-APar, which we could not test due to its unavailability) of this second pathway further strengthens the existence of enzyme system necessary for its utilization, in this bacterium. Currently, we do not understand how these two possible pathways operate in this bacterium and how these might be regulated.

To correlate a relationship between degradation and chemotaxis, we have also monitored the chemotactic behavior of Pseudomonas sp. strain BUR11 toward parathion, chlorpyrifos and their major degradation intermediates such as 4-NP, 4-AP and TCP using qualitative and quantitative chemotaxis assays. The strain BUR11 showed chemotaxis towards parathion, chlorpyrifos and their degraded intermediate products 4-NP, 4-AP as well as TCP. Although genetic analysis is needed for better understanding of this chemotactic process, nevertheless, this feature is novel and unique to the best of our knowledge. Although mineralization of chlorpyrifos and TCP was not addressed in this study, the capacity of the strain BUR11 to utilize them as the sole source of carbon for growth strongly suggests that these compounds might also be mineralized. Thus, the strain BUR11 showed chemotaxis towards those compounds which it can metabolize (i.e., utilize and degrade chemo-attractants). However, we are not sure whether this chemotactic response is metabolism dependent or independent (Pandey & Jain, 2002), as the chemotactic response of the BUR11 was not studied for those compounds which it cannot degrade or metabolize. Metabolism dependent chemotaxis towards other toxic xenobiotic compounds has been reported for many bacteria, such as Burkholderia sp. SJ98 towards various nitroaromatic compounds (Pandey et al., 2012), Pseudomonas sp. JHN towards 4-chloro, 2-NP (Arora & Bae, 2014) and Ralstonia eutropha towards 2, 4-dichlorophenoxyacetate (Hawkins & Harwood, 2002). Unlike these when chemotaxis occurs independent of metabolism of chemo-attractants, it is called metabolism independent chemotaxis as has been reported for Pseudomonas sp. WBC-3 and Pseudomonas putida PRS2000, towards various aromatic compounds (Zhang et al., 2010; Parales, 2004; Harwood & Ornston, 1984; Harwood, Parales & Dispensa, 1990). Chemotaxis offers a selective advantage to degrading bacteria by guiding them to sense and locate toxic xenobiotic pollutants that might be present in the environment. Chemotaxis towards toxic pollutants and their mineralization indicate that chemotaxis might be an integral feature of degradation (Pandey & Jain, 2002; Marx & Aitken, 2000) and its proper understanding might offer development of better bioremediation strategies for OP contaminated sites.

Genes for degradation of OP compounds are reported to be present in broad host range plasmids such as pMCP424 and this has been shown to be disseminated among other soil dwelling microorganism by transformation, conjugation and transduction (Horne et al., 2002; Bhadbhade, Sarnaik & Kanekar, 2002; Ochman, Lawrence & Groisman, 2000; Springael & Top, 2004). However, no plasmids could be detected in strain BUR11. Moreover, plasmid curing treatments also did not change the growth potential and vigor on OP compounds for the same strain, thereby indicating, possibility of chromosome encoded genetic component to be involved in degradation.

Due to its indiscriminate, widespread, and persistent use, especially in India, OP insecticides are a growing health issue that requires public attention and awareness. Although several chemical and physical methods have been developed for removal of OP compounds (Theriot & Grunden, 2011) from its contaminated sites, these are technically and economically challenging (Megharaj et al., 2011). Bioremediation process involving living microorganisms or their enzymes has recently received tremendous attention as eco-friendly, cost effective, safer approach to clean up contaminated environments (Singh, 2009; Chen et al., 2011; Chen et al., 2012b). Coordinated studies carried out by addressing chemotaxis and degradation studies of toxic OP compounds might develop better integrated approach towards bioremediation of these compounds. In most cases, the failure of bio-augmentation approaches is due to the inability of the microorganisms to tolerate biotic and abiotic stresses and application of microorganisms with better stress tolerance capacity might improve these approaches (Tyagi, da Fonseca & de Carvalho, 2011). In this regard, chemotactic, thermo-halo tolerant bacterium, Pseudomonas sp. BUR11 with OP degradation activity have better bioremediation capacity and might have immense biotechnological potential for cost effective cleanup of OP contaminated sites.

Conclusions

Pseudomonas sp. BUR11 can degrade parathion using two pathways. The first is through the formation of 4-NP while the second pathway is through the formation of 4-APar and 4-AP. The strain also showed positive chemotaxis response towards the OP compound, 4-NP, 4-AP and TCP. This is the first report of chemotaxis towards the OP compound by a thermo- halo tolerant bacterium.

Supplemental Information

Figure S1 Comparative growth profile of BUR11 on increasing concentration of NaCl

Click here for additional data file.

Figure S2 Drop plate assay and Swarm plate assay

Qualitative chemotactic response of BUR11 towards parathion, chlorpyrifos, 4-NP, 4-AP and TCP. (A) Drop plate assay and (B) Swarm plate assay.

Click here for additional data file.

Figure S3 TLC analysis

Identification of metabolites of parathion degradation by TLC. Authentic standards 1, parathion; 2, 4-NP; 3, PBQ; 4, HQ; 5, 4-NC; 6, BT; 7, 4-AP. While, 8 and 9 correspond to 72 h and 120 h extract of parathion grown culture, indicating the detection of 4-NP, PBQ and BT during course of degradation.

Click here for additional data file.

Figure S4 HPLC chromatograms

Parathion degradation by strain BUR11. The elution profile of each sample is shown as individual HPLC chromatograms. 0 h test sample (A), 24 h test sample (B) and 96 h test sample (C) and elution profile of standards [D, Parathion; E, 4-NP; F, PBQ; G, HQ; H, 4-NC and I, BT].

Click here for additional data file.

Figure S5 GC chromatogram of parathion grown culture

GC spectra of parathion grown culture. The peak having RT of 19.030 min was identified as 4-APar by NIST library search.

Click here for additional data file.

Figure S6 LC-MS/MS spectra of major intermediates of parathion

LC-MS/MS spectra of major intermediates of parathion degradation by strain BUR11. A, 4-NP; B, HQ; C, 4-AP; D, PBQ; E, 4-NC and F, BT. Metabolites were identified and confirmed based on matches from NIST library search.

Click here for additional data file.

Supplemental Information 1 Raw data

Click here for additional data file.

We are grateful to BUREAU VERITES, Chenni for GC and LC-MS/MS; Dr W Ghosh, Microbiology Dept., and Mr Swaroop Biswas, CIF, Bose Institute, Kolkata, for their help in HPLC; Dr Tapan Chakrabarti, MCC, NCCS, Pune for useful scientific discussions and manuscript correction.

Additional Information and Declarations

Competing Interests

Author Contributions

DNA Deposition

The authors declare there are no competing interests.

Santanu Pailan performed the experiments, analyzed the data, prepared figures and/or tables, reviewed drafts of the paper.

Pradipta Saha conceived and designed the experiments, analyzed the data, contributed reagents/materials/analysis tools, wrote the paper, reviewed drafts of the paper.

The following information was supplied regarding the deposition of DNA sequences:

GenBank: KF887018.

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
