# Peer review of "Chemotaxis and degradation of organophosphate compound by a novel moderately thermo-halo tolerant Pseudomonas sp. strain BUR11: evidence for possible existence of two pathways for degradation"

_PeerJ, doi:10.7717/peerj.1378_

## Round 0.1 · original submission · Major Revisions

1. English should be improved.
2. Kindly include pictures of drop plate and swarm plate assays.
3. Discussion section should be improved and more references should be added.
4. Kindly check the possibility of plasmid coded pathway.

I look forward to your response to the reviewers.

·

Basic reporting

In this manuscript entitled “Chemotaxis and degradation of organophosphate compound by a novel moderately thermo-halo tolerant Pseudomonas aeruginosa strain BUR11: evidence for existence of two pathways for degradation”, the authors isolated a organophosphate-degrading bacterium Pseudomonas aeruginosa BUR11. The degradation abilities of strain BUR11 were studied, and the intermediates of parathion degradation were investigated. Moreover, the authors reported for the first time that the bacterial metabolism depended chemotaxis towards organophosphate compounds. Overall, the manuscript provides some interesting information; the results are clear except a few weak points.

Experimental design

In general, this paper is well planned and easy to read. The experiments are well designed and appropriate controls are presented.

Validity of the findings

The data presented in this manuscript will be of great interest to the readers and extend to a high degree our knowledge about organophosphate compound behaviour.

Additional comments

In this manuscript entitled “Chemotaxis and degradation of organophosphate compound by a novel moderately thermo-halo tolerant Pseudomonas aeruginosa strain BUR11: evidence for existence of two pathways for degradation”, the authors isolated a organophosphate-degrading bacterium Pseudomonas aeruginosa BUR11. Below are the points that this reviewer has noticed.

1. Introduction needs to be completed. A more through description of the environmental fate characteristics of parathion is needed based on the literature available and described in original paper. This way the authors will demonstrate that they really have a good knowledge of the related literature. How far it degrades (half-life)? The authors need to add this information in the first paragraph. I also propose complete this section with more physico-chemical information about parathion, e.g. volatility, degradability, fate in various environments, etc.

2. Line 42: “Organophosphate (OP) compounds are extensively used throughout the world as insecticides, pesticides as well as chemical warfare agents... ”, this sentence is comfusing. Pesticides include insecticides. Use “nematicide” instead of “pesticides” here.

3. Line 52: “Reports in the literature suggest that few soil microorganisms... ” this is an inexact description.
Actually, a number of reports on degradation of chlorpyrifos and its intermediate by bacterial or fungal species have already been available in literatures. Such as:
Biodegradation of chlorpyrifos and its hydrolysis product 3,5,6-trichloro-2-pyridinol by a new fungal strain Cladosporium cladosporioides Hu-01. PLoS One, 7: e47205. (2012)
Development of a freeze-dried fungal wettable powder preparation able to biodegrade chlorpyrifos on vegetables. PLoS One, 9: e103558. (2014)
Plasmid-mediated bioaugmentation for the degradation of chlorpyrifos in soil. J Hazard Mater 221-222: 178-184. (2012)
Purification and characterization of a novel chlorpyrifos hydrolase from Cladosporium cladosporioides Hu-01. PLoS One 7: e38137. (2012)
Isolation and characterization of 3,5,6-trichloro-2-pyridinol-degrading Ralstonia sp. strain T6. Bioresour Technol 101: 7479-7483. (2010)
The author need to incorporate these related references in this part, and use them to develop an interesting discussion which could supplement to this study on biodegradation processes carried out by environmental microorganisms.
In addition, the information about enzymes or genes involved in the degradation of chlorpyrifos should be provided in this part.

4. Line 73: Abstract: “ has”, should be “have” ?

5. Line 87: “HPLC”, please use the full name for the first time. The same as the followings “ EDTA”, “ TLC, LC-MS/MS and GC”.

6. Line 93: “ MgSO 4. 7H 2 O”, “ CaSO 4. H 2 O”, “ NaMoO 4. 2H 2 O”, “FeSO 4. 7H 2 O”, the comma symbol should be in the middle.

7. Line 138: delete “100 mM potassium phosphate, pH 7.0 and 20 µM EDTA”.

8. Line 243: “LC-MS-MS”, use “LC-MS/MS”.

9. References: Many of the references have been superceded and more modern ones are required.

Reviewer 2 ·

Basic reporting

The manuscript entitled “Chemotaxis and degradation of organophosphate compound by a novel moderately thermo-halo tolerant Pseudomonas aeruginosa strain BUR11: evidence for existence of two pathways for degradation” represents a well planned and executed study on an important and pertinent aspect of environmental biotechnology. The manuscript has been prepared meticulously and reads quite well for scientific content as well as presentation.

Authors report isolation and characterization of a bacterial strain capable of chemotaxis and degradation of parathion (an important organophosphate compound). In addition to the characterization of chemotactic behavior of isolated strain towards parathion, authors have also elucidated the catabolic pathway for its degradation. Authors have also presented preliminary evidence showing that the isolated bacterial strain is capable of degrading parathion via 2 inter-connected pathways.

Although, a large number of studies have been carried out and reported on bacterial chemotaxis and degradation of toxic xenobiotic pollutants, yet a study focusing on bacterial chemotaxis towards organophosphate compounds are extremely scarce. Therefore, this manuscript certainly holds merit. However, there are a few concerns that must be addressed by the authors for ensuring the highest quality of the manuscript before acceptance for publication in Peer J.

Major Comments:

Abstract:

P2, L33- 34: Authors have shown that the newly isolated strain viz., Pseudomonas aeruginosa BUR11 degrades parathion via 2 pathways one of which includes 4-NP as a major identified intermediate. Since 4-NP is a well-established bacterial chemoattractant, It is possible that strain BUR11 is exhibiting chemotaxis towards the 4-NP (generated during degradation) rather than parathion itself. Further, the graph showing quantitative chemotactic response of strain BUR11 also indicates very similar concentration response towards Parathion and 4-NP.

Therefore, at this point it is very difficult to conclude that strain BUR11 is certainly exhibiting chemotaxis towards parathion. To categorically demonstrate chemotaxis towards parathion, a mutant deficient in parathion to 4-NP conversion would be required.

Additionally, in a discreet report, Wen Y et al., 2007 (Wei Sheng Wu Xue Bao. 2007 Jun;47(3):471-6.) have previously reported isolation and characterization of Pseudomonas putida strain DLL-1 for degradation and chemotaxis to Methyl Parathion, which is also an organophosphate.

P2, L27- 33: Experiments carried out during the present study and results presented in the manuscript may not be sufficient for concluding that there are 2 fully operational catabolic pathways are present in strain BUR11 for metabolism of parathion. To reach to such conclusion other experiments e.g. gene- characterization, enzymatic assays and selective gene deletion mutation would be required.

Still the results presented suggest about possibility of co-existence of 2 pathways. Therefore, at this point of the study, authors may try to mellow down this conclusion.

Materials and Methods
It would be useful if the concentrations of different substrates/ chemoattractant used during the present study are represented as ppm or micro/ milli molar concentrations. It would be much easier to comprehend.

Results:
P10, L199- 200: It is quite interesting that a xenobiotic compound degrading organism is also halo- tolerant in nature. However, the result from experiments carried out for growth of BUR11 on NaCl has been under represented in the manuscript. A comparative growth profile of BUR11 on increasing concentration of NaCl must be worth mentioning and will interest general audience.

P11, L215: Authors have mentioned that they performed both qualitative and quantitative chemotaxis assays, however, in the manuscript; results are presented for only the quantitative chemotaxis assay. It is equally important to include the figures from drop plate and swarm plate assays.

P11, L231- 235: From Fig. S2B the concentration of HQ (Rt = 2.25) and from Figure S2C, the concentration of 1,2,4- BT (Rt = 2.196) seems to be great than that of 4-NP at time 24 Hrs and 96 Hrs respectively. Authors should re-visit their conclusion that “HQ, PBQ and BT were detected in very low concentrations.
Discussion:

P14, L295- 298: Authors have shown that strain BUR11 exhibits chemotaxis towards compounds that it can metabolize. In the very next line they mention that “It is called metabolism dependent chemotaxis”. In my opinion, the metabolism dependent chemotaxis is one where chemotaxis cannot occur in absence of the metabolism.
Although the manuscript has clearly shown chemotaxis of strain BUR11 towards the compounds that it metabolizes, yet it is not yet certain if it can occur only when metabolism takes place.

Authors should re-visit this statement in the discussion section.

Minor Comments:

Introduction:

P4, L69: Please quote an appropriate reference for the statement “chemotaxis is considered important in biofilm formation and guide bacterium to swim towards nutrients.

P4, L70-72: The statement “These aspects are currently interesting topic of research to environmental microbiologists” should be amended considering the references quoted (e.g. O’Toole & Kolter 1998 etc.) are ~ 17 years old.

Materials and Methods

Chemicals

P5, L85- 88: Please check to consistency with respect to reporting of the name of manufacturer, distributor, or vendor all along the document.
Characterization and identification of isolate BUR11

P6, L 108: Please check statement “Strain MemC14 was also identified by automated …..

Growth studies and chemotactic behavior of strain BUR11

P6, L125: Author may consider replacing ‘toxic compound’ with another word e.g. chemoattractant because in my opinion, none of the tested compounds were toxic to BUR11.

Analytical Methods

Quantification of residual parathion…….

P8, L168- 170: Whether the HLPC program used was as per any previous reference of it was developed de novo. Authors may consider mentioning accordingly. Similarly for Identification of metabolites by TLC, HPLC, LC-MSMS and GC, it is not clear if authors have used a previously reported protocol or developed their own protocol. It would be useful for the readers if this information is included.

P12, L244- 245: The mass spectra for BT and 4-NC shown in figure S4E and S4F respectively look identical. Authors are recommended to check for accuracy of used figures at their end.

Experimental design

The experimental design used in the present study/ manuscript is scientifically sound. However, a few more experiments would be required to confirm some of the conclusions drawn by the authors.

Validity of the findings

Most of the finding reported in the manuscript have been validated by more than one experimental methods. Therefore, the findings could considered validated. Again a few more experiments would be necessary to conclude some of the observations.

Additional comments

The research study reported in the current manuscript represents a well planned and executed study on an important and pertinent aspect of environmental biotechnology. The manuscript has been prepared meticulously and reads quite well for scientific content as well as presentation.

However, some of the conclusions (e.g. co- existence of 2 catabolic pathways etc.) need to be validated with supporting experiments.

---

## Round 0.2 · Minor Revisions

1. Line 22-23, :A novel organophosphate (OP) degrading chemotactic bacterial strain was isolated and identified
23 as Pseudomonas aeruginosa BUR11" should be replaced by " An organophosphate (OP) degrading chemotactic bacterial strain BUR11 isolated from an agricultural field was identified as a member of Pseudomonas genus on the basis of its 16S rRNA gene sequence.

2. Kindly use Pseudomonas sp. rather than Pseudomonas aeruginosa throughout the manuscript because you have not characterized strain up to species level.

3. Kindly correct LC-MSMS by "LC-MS/MS" throughout the manuscript.

4. Kindly recheck optical density of bacteria in presence of pesticides. It is too high. Kindly repeat and confirm.

5. How did you identify LC-MS/MS chromatograms. NIST library or other library. Kindly include number of library.

6. Kindly read and proof manuscript for other technical errors.

Reviewer 2 ·

Basic reporting

It is quite appreciable that Authors have taken great care in addressing the concerns raised during revision of the manuscript. The revised manuscript is significantly improved. According to my opinion, it is now suitable for publication in a Peer J.

Experimental design

The experimental design, result interpretation, data analyses and presentation is suitable for publication.

Validity of the findings

Authors have adequately validated their findings with use of complementary experimentation. All of the findings presented are well validated in the revised manuscript.

Additional comments

The study represents a very interesting research work and opens up avenue for further exciting studies (e.g. molecular and biochemical characterization, whole genome analyses, chemo-attractant signaling etc.).

Regarding the manuscript, in my honest opinion, the revised version is significantly better than the earlier version. (Thanks for being pro-active to constructive criticism and addressing most of the concerns positively).

---

## Round 0.3 · accepted · Accept

The manuscript is suitable for publication.